# Capsaicin: A Potential Treatment to Improve Cerebrovascular Function and Cognition in Obesity and Ageing

**DOI:** 10.3390/nu15061537

**Published:** 2023-03-22

**Authors:** Tammy Thornton, Dean Mills, Edward Bliss

**Affiliations:** 1School of Health and Medical Sciences, University of Southern Queensland, Ipswich, QLD 4305, Australia; 2Respiratory and Exercise Physiology Research Group, School of Health and Medical Sciences, University of Southern Queensland, Ipswich, QLD 4305, Australia; 3Centre for Health Research, Institute for Resilient Regions, University of Southern Queensland, Ipswich, QLD 4305, Australia; 4Molecular Biomarkers Research Group, University of Southern Queensland, Toowoomba, QLD 4350, Australia

**Keywords:** capsaicin, cerebrovascular, cognition, obesity, ageing

## Abstract

Impaired cognition is the primary symptom of dementia, which can lead to functional disability and reduced quality of life among an increasingly ageing population. Ageing is associated with increased oxidative stress, chronic low-grade systemic inflammation, and endothelial dysfunction, which reduces cerebrovascular function leading to cognitive decline. Chronic low-grade systemic inflammatory conditions, such as obesity, exacerbate this decline beyond normal ageing and predispose individuals to neurodegenerative diseases, such as dementia. Capsaicin, the major pungent molecule of chilli, has recently demonstrated improvements in cognition in animal models via activation of the transient receptor potential vanilloid channel 1 (TRPV1). Capsaicin-induced TRPV1 activation reduces adiposity, chronic low-grade systemic inflammation, and oxidative stress, as well as improves endothelial function, all of which are associated with cerebrovascular function and cognition. This review examines the current literature on capsaicin and Capsimax, a capsaicin supplement associated with reduced gastrointestinal irritation compared to capsaicin. Acute and chronic capsaicin treatment can improve cognition in animals. However, studies adequately assessing the effects of capsaicin on cerebrovascular function, and cognition in humans do not exist. Capsimax may be a potentially safe therapeutic intervention for future clinical trials testing the effects of capsaicin on cerebrovascular function and cognition.

## 1. Introduction

Dementia affects approximately 50 million people worldwide and is expected to rise to 130 million by 2050 [1]. Dementia is a major cause of functional disability, with a substantial social and economic burden estimated to cost USD 2.8 trillion globally by 2030 [1,2,3]. The primary symptom of dementia is impaired cognition, which is characterised by a reduction in cognitive functions including memory, language, thinking, and/or judgment [4]. Reductions in these functions cause psychological changes that impact social relationships and independence, which ultimately affect quality of life (QoL) (Figure 1) [5,6]. The reduced cognition observed in dementia is typically preceded by a reduction in cerebrovascular function [7,8].

Obesity and ageing lead to chronic low-grade systemic inflammation and oxidative stress, which predisposes an individual to the development of dementia [9,10,11]. Obesity is a current epidemic that affects two billion people worldwide and is defined as an excessive accumulation of fat that may impair health [12]. The number of overweight and obese people worldwide has tripled since 1975 [13,14,15]. Similarly, the proportion of people aged over 60 years will nearly double from 12% in 2015 to 22% in 2050, with people living longer than previously [16].

The current treatments for dementia are costly, have significant side effects, and tend to target symptoms of cognitive decline in isolation. Further, there is also no clear evidence that pharmaceuticals or cognitive training programs are effective in preventing or stopping cognitive decline [17,18]. Prevention strategies such as cognitive training, physical activity and dietary and lifestyle changes can also have low compliance, particularly for those already suffering from cognitive decline [18,19]. Therefore, new treatment strategies are urgently needed. Novel nutraceutical interventions hold promise as complementary approaches to prevent cognitive decline [20]. Further, there is a need for research on multi-functional treatments that address a variety of factors that can lead to cognitive decline [18,21].

Capsaicin, a pungent molecule found in plants belonging to the *Capsicum* genus, has been shown to have neuroprotective and anti-inflammatory properties in animal models [22,23,24,25]. In animal studies, capsaicin decreases chronic low-grade systemic inflammation and oxidative stress, which may ultimately improve cerebrovascular function and reduce the symptoms of cognitive decline and dementia [10,26,27]. Therefore, the aims of this review are to (1) provide a brief overview of cognition and cerebrovascular function; (2) examine the mechanisms underlying how increased adiposity and ageing lead to reduced cerebrovascular function and cognition; (3) examine how capsaicin attenuates the effects of ageing and obesity on decreased cerebrovascular function and cognition; and (4) provide a summary of potential health benefits associated with Capsimax, a capsaicin supplement associated with reduced gastrointestinal irritation. 

## 2. An Overview of Cognition and Cerebrovascular Function

Cognition, which is one of the brain’s primary functions, is the mental processing that occurs when acquiring, encoding, adapting, and applying sensory experiences from the environment [28,29]. This encompasses perception, reasoning, awareness, and comprehension [29]. Cognition depends not only on the ability of the brain to acquire information via sensory input but it also depends on the processing and integration of this information, which guides behaviour and actions [28]. A reduction in these processes results in mood and behavioural disturbances, social disruptions, loss of independence, and increased care, ultimately leading to a lowered QoL [5,6,30].

The brain is one of the most metabolically active organs, consuming 15–20% of the body’s nutrients and energy under resting conditions [31]. The brain cannot store nutrients and oxygen and, therefore, requires a constant supply of these through the cerebral blood flow (CBF) to function optimally and maintain its primary functions, particularly cognition [32]. This supply is maintained by the function of the cerebrovasculature.

Cerebrovascular function describes the ability of the cerebrovasculature to perfuse the brain with adequate blood (i.e., CBF) in response to physical and psychological stimuli [33,34]. This is achieved through neurovascular coupling (NVC) and cerebral autoregulation [33]. These are the complex mechanisms that ensure a chemically stable environment in response to increased neuronal metabolism and environmental, chemical and/or mechanical (i.e., physical) changes [2,29,33]. Changes in cerebral blood pressure, perfusion pressure, vascular diameter, and blood viscosity all affect CBF [29,35,36]. Autoregulation ensures that a constant mean arterial pressure of 50–160 mmHg is maintained in response to chemical and/or mechanical stimuli [33,34]. This response is largely initiated by either increased (vasodilatation) or decreased (vasoconstriction) nitric oxide (NO) metabolism, which regulates the resistance applied globally to the cerebrovasculature [32,37]. Conversely, NVC is where neurons communicate with endothelial cells to release vasodilatory mediators, such as NO, to maintain CBF locally during periods of increased neuronal metabolism [37,38].

The primary contributing factor maintaining cerebrovascular function, and therefore CBF, is endothelial function which is fundamental to cerebrovascular autoregulation and NVC [39]. The upregulation of eNOS catalyses the synthesis and release of NO [32,37,40]. NO is released into vascular smooth muscle, causing rapid and sustained vasodilatation and, therefore, increased blood flow [37]. Additionally, NO regulates inflammation and oxidative stress via multiple pathways, thereby reducing chronic low-grade systemic inflammation [41,42]. This is important because when NO production is reduced, it can cause changes in CBF regulation, which leads to vascular insults [10,29,43]. This results in reduced oxygen and nutrient delivery and structural and functional changes in the brain, which precede cognitive decline and dementia [10,29,43]. This process is evidenced in studies that have indicated cerebrovascular dysfunction is the second leading cause of Alzheimer’s disease (AD) and the most common cause of vascular dementia, which are arguably the two most prevalent forms of dementia [7]. It is therefore important to be able to readily measure cerebrovascular function together with cognition.

## 3. Ageing and Obesity Are Risk Factors for Cerebrovascular Dysregulation

Ageing and increased adiposity have synergistic detrimental effects on endothelial function, due to hormonal changes, chronic low-grade systemic inflammation and increased reactive oxygen species (ROS) production (Figure 2) [44,45]. Uncoupling of eNOS creates ROS instead of NO, causing oxidative stress which exacerbates endothelial dysfunction [32]. This results in impaired CBF, leading to hypoperfusion that causes structural and functional changes in the brain which precede cognitive decline [44,45]. Endothelial NO is, therefore, one of the most important signalling molecules for maintaining CBF through autoregulation and NVC, and a potential target for preventing cerebrovascular and neurodegenerative diseases [39,46].

### 3.1. Ageing

Life expectancy has increased compared to previous generations [16]. This is important because ageing is the greatest predictor of cognitive decline, with the most significant impact on neurodegeneration [44,47,48]. Molecular and cellular damage accumulated over time leads to structural and functional brain changes, impaired cognition, and neurodegeneration [47,49]. Ageing is also a key factor leading to endothelial dysfunction and cerebrovascular dysregulation [42,50,51].

Ageing elevates circulating endothelial NO synthase (eNOS) inhibitors, such as asymmetrical dimethylarginine and arginase, uncoupling eNOS and preventing the vital NO synthesis cofactor, L-arginine, from being metabolised [52]. This, in turn, reduces NO synthesis and bioavailability [52,53,54]. Conversely, insulin-like growth factor-1 (IGF-1) and brain-derived neurotrophic factor (BDNF), which stimulate NO production are decreased with age [32,55]. Further, BDNF plays a role in neuron survival, growth and maintenance [56]. IGF-1 is important for neurotransmitter synthesis and supports BDNF production [57]. The reduction BDNF has therefore been associated with Alzheimer’s disease pathology [58]. Tetrahydrobiopterin (BH4) is an essential cofactor for NO production. BH4 also decreases with age due to increased oxidation and decreased synthesis, as well as age-related changes in metabolism, such as reduced folate and B12 absorption [59,60]. Decreased BH4 leads to eNOS uncoupling and production of ROS instead of NO production [61]. Microglia are specialised immune cells in the brain which contribute to synaptic plasticity [62]. Microglia assist with clearing misfolded proteins or damaged cells resulting from neurodegeneration [63]. However, increased age causes impaired and prolonged activation of microglia, which increases pro-inflammatory mediators and ROS, exacerbating neuronal damage [64,65,66]. Additionally, cytokines and low shear stress stimulate the release of endothelins from endothelial cells, particularly ET-1, which is a powerful vasoconstrictor [67]. The increase of ET-1 initiates increased expression of transcription factors that promote inflammation, particularly nuclear factor kappa B (NF-ĸB) [68]. NO inhibits ET-1. Therefore, the reduction of NO due to increased ET-1 leads to an imbalanced ET-1 and NO ratio [69]. This also leads to increased ROS and inflammation, resulting in endothelial dysfunction [32,70]. Endothelial dysfunction causes decreased cerebrovascular function and reduced CBF which precedes cognitive decline [71]. Further to the natural progression of endothelial dysfunction seen with ageing, obesity exacerbates endothelial dysfunction and has also been associated with the risk of dementia [72].

### 3.2. Obesity as a Risk Factor for Cognitive Decline

Obesity is a preventable and modifiable metabolic disease associated with chronic low-grade systemic inflammation, increased oxidative stress and endothelial dysfunction, which decreases CBF and promotes cognitive impairment (Figure 2) [73]. Increased adipose tissue is the key feature of obesity [74]. Adipose tissue secretes over 600 signalling molecules, called adipokines [75]. Increased adiposity reduces the production of anti-inflammatory adipokines, such as adiponectin, and increases inflammatory adipokines, such as leptin, therefore increasing inflammation [76]. Adiponectin is downregulated by inflammatory molecules, and oxidative stress, further promoting systemic inflammation [77]. Overexpression of inflammatory cytokines stimulates the attachment of leukocytes to the endothelium. This results in increased vascular permeability, occlusions and inflammation, thereby reducing eNOS function, systemic blood flow and CBF [78]. Therefore, increased adiposity is a risk factor for cardiovascular disease, which correlates with reduced CBF, hypoperfusion and cognitive decline [11,79]. A meta-analysis of seven observational studies demonstrated that obesity increased the risk of dementia between 32 and 45% due to increased concentrations of inflammatory markers [80]. Therefore, dementia research including populations with increased adiposity is required [81]. Nutraceuticals, such as capsaicin, have demonstrated promising effects in reducing the detrimental effects of adiposity on endothelial dysfunction, adiposity itself, and cardiovascular disease [82,83]. In animal models, capsaicin has been shown to reduce the release of obesity-related pro-inflammatory cytokines IL-6, and tumour necrosis factor-α (TNF-α), therefore reducing inflammation, improving eNOS function, NO synthesis, cerebrovascular function and cognition [27]. Although these results are clear in animal models, little research has been conducted on humans to determine the effects of capsaicin specifically on cerebrovascular function or cognition.

**Figure 2 nutrients-15-01537-f002:**
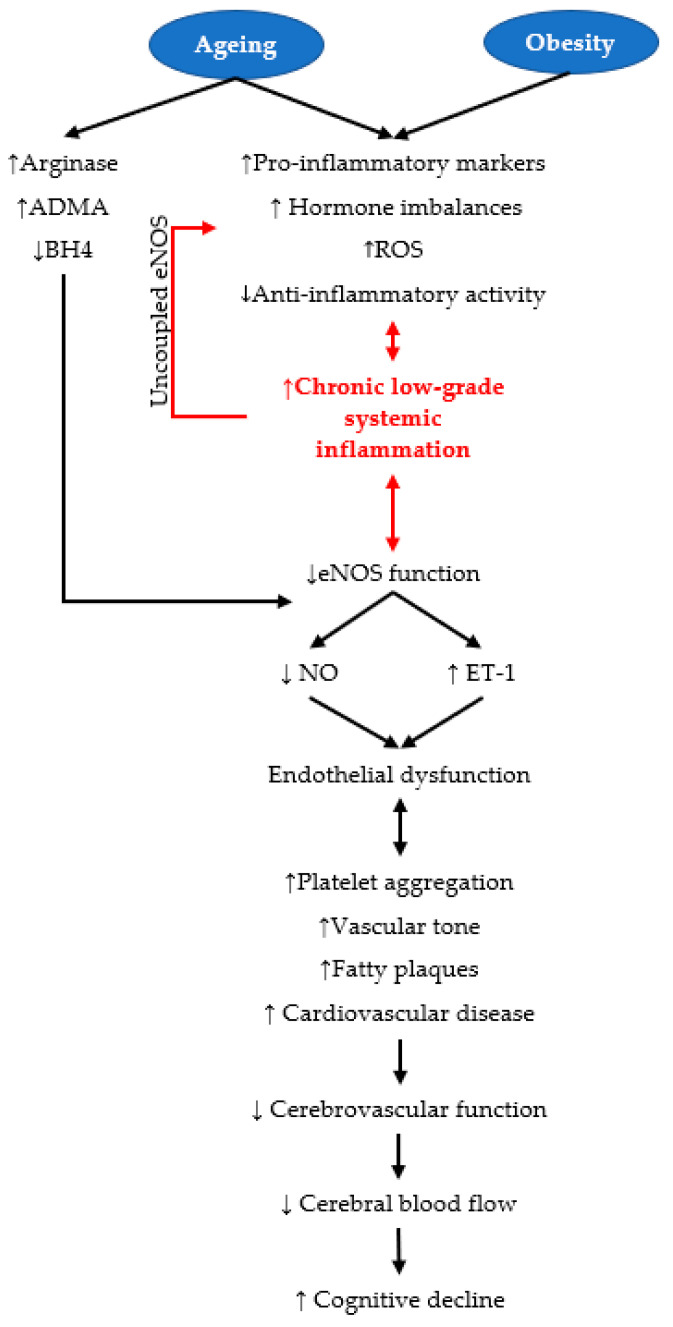
The effect of ageing and obesity on endothelial dysfunction, leading to cognitive decline. Obesity and ageing increase the production of inflammatory markers, including tumour necrosis factor-alpha (TNF-α), and interleukin-6 (IL-6) and decrease anti-inflammatory mediators [66,76]. Increased reactive oxygen species (ROS) occurs in obesity and ageing, further promoting chronic low-grade systemic inflammation [68,76,84]. Ageing also reduces the endothelial nitric oxide synthase (eNOS) cofactor, tetrahydrobiopterin (BH4), and increases asymmetrical dimethylarginine (ADMA), which decreases nitric oxide (NO) synthesis and bioavailability [68]. Increased arginase activity reduces L-arginine supply, thus also promoting the uncoupling of eNOS [53]. Increased vasoconstrictors, such as endothelin-1 (ET-1), contribute further to increased vascular tone and endothelial dysfunction [67]. Uncoupled eNOS creates further ROS instead of NO, exacerbating endothelial dysfunction [61]. These mechanisms lead to low-grade systemic inflammation and ED, leading to increased platelet aggregation, vascular tone and fatty plaques leading to cerebrovascular dysfunction, thus reducing cerebral blood flow and cognition [85]. Abbreviations: ↑: increased; ↓: decreased.

## 4. Capsaicin

The capsaicinoids are the primary pungent molecule found in plants belonging to the *Capsicum* genus, particularly chillies [86]. They are a group of phenolic compounds characterised by a common vanilloid ring (Figure 3) [87]. Capsaicin, dihydrocapsaicin, nordihydrocapsaicin, homocapsaicin and homodihydrocapsaicin all comprise the capsaicinoids, with capsaicin being the most abundantly occurring of these [87,88]. Capsaicin has been extensively studied for its multiple benefits as an anti-carcinogenic, anti-inflammatory, antioxidant, and anti-obesity agent and for its use as a topical analgesic [89,90]. Table 1 summarises research studies that have investigated these effects.

When taken orally, capsaicin is passively absorbed in the stomach and jejunum [26,87]. Albumin transports capsaicin in the blood, and its vanilloid ring has a high affinity for the transient receptor vanilloid 1 (TRPV1), a non-selective cation channel [87,91]. TRPV1 receptors are widely expressed in the body and are found to be concentrated in neural tissue (peripheral and central) and the endothelium [83]. When capsaicin binds to TRPV1, it causes a cation influx, activating numerous physiological pathways which are important modulators of vasodilation and inflammation (Figure 4) [92,93,94]. In turn, downregulation of these pro-inflammatory pathways and upregulation of anti-inflammatory pathways promote the increased expression of eNOS and, therefore, increased NO production and availability, counteracting the effects of endothelial dysfunction [93,95,96].

**Table 1 nutrients-15-01537-t001:** Summary of studies outlining the effects of capsaicin on chronic disease states.

Disease State	Main Findings	Reference
Cardiovascular	↓Blood pressure	[97]
Cancer	Anti-proliferative	[88]
Neuropathic pain	↓Painful neuropathy	[98,99]
Adiposity and metabolic derangements	↑Energy expenditure ↑Fat oxidation ↑Thermogenesis ↑Glucose tolerance ↑Insulin sensitivity ↑Resting metabolic rate ↓Body mass ↓Total cholesterol ↓Triglycerides ↓ Glucose	[83,100,101,102,103,104]

Abbreviations: ↑: increased; ↓: decreased.

### Capsaicin: A Brief Overview of Its Role as Anti-Cardiometabolic Disease Treatment

Capsaicin’s action in reducing obesity, oxidative stress, and inflammation and improving cardiovascular function in animals has been previously described [83,86,105]. Obesity increases blood triglycerides, free fatty acids and low-density lipoprotein (LDL), contributing to eNOS dysregulation, vascular remodelling, and atherosclerosis, which is the leading cause of cardiovascular and cerebrovascular disease [83,93,106].

Capsaicin increases thermogenesis, energy expenditure and fat oxidation, all of which assist in decreasing adiposity [83,107]. This is achieved by the activation of TRPV1 and subsequent reduction of pro-inflammatory cytokines, such as TNF-α and IL-6, which are increased with greater adiposity [27,78,108]. Ma et al. [109] found that dietary capsaicin (0.1%) for 24 weeks in C57BL/6J mice activated TRPV1, inducing cytosolic calcium and reduced lipid accumulation and atherosclerosis [106]. Wang, Y. et al. [110] cultured human umbilical cord endothelial cells and treated them with capsaicin. Capsaicin increased NO, and reduced cytokine production, monocyte adhesion, adhesion molecule expression and activated NF-ĸB, thereby reducing inflammation. TRPV1 activation with 1 µM capsaicin rescued impaired macrophage autophagy induced by oxidised low-density lipoprotein, activating AMPK signalling, inhibiting foam cell formation, and preventing atherosclerotic plaque formation [106]. Dai et al. [111] fed male apolipoprotein E knock-out mice 0.01% capsaicin for 18 weeks alongside a high-fat diet or a high-fat diet with broad-spectrum antibiotics. Capsaicin reduced serum lipopolysaccharide (an inflammatory mediator) and low-density lipoprotein, as well as increased high-density lipoprotein. This was not observed in the group fed capsaicin with concurrent antibiotics. The improvements resulted in reduced intestinal inflammation and permeability, as well as improved endothelial function, which led to a significant reduction in atherosclerotic lesions. This demonstrates capsaicin’s ability to reduce adiposity and its associated inflammatory mediators, as well as reducing cardiovascular-induced risk factors that can reduce cerebrovascular function and cognition. This may also indicate capsaicin’s application as a resolution to other inflammatory diseases such as autoimmune diseases, gastrointestinal disorders, and other haemodynamic conditions, such as atherosclerosis.

**Figure 3 nutrients-15-01537-f003:**
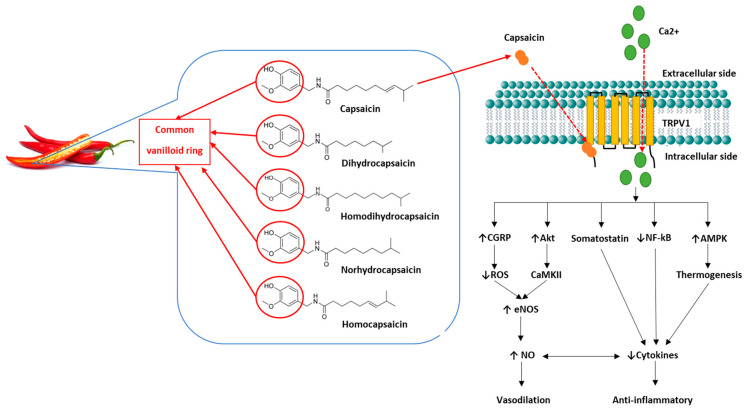
Capsaicin structure and function. Capsaicin and dihydrocapsaicin contribute the most to the pungency of chilli and are found primarily in the fruit pod [86]. The capsaicinoids are characterised by their common vanilloid ring (circled). Capsaicin consists of a *trans* configuration with a double-bond and an even number of branched-chain fatty acid moieties [89]. The vanilloid ring binds intracellularly to TRPV1 channels on cell membranes, causing an influx of extracellular calcium into the cell and triggering numerous physiological pathways [82,94]. Capsaicin causes the release of sensory neuropeptides such as calcitonin gene-related peptide (CGRP) [112]. CGRP is a potent vasodilator that reduces reactive oxygen species (ROS) production by promoting increased endothelial nitric oxide synthase (eNOS) function and nitric oxide (NO) production [94,112]. Somatostatin reduces pro-inflammatory cytokines, inducing anti-inflammatory and immunomodulatory effects [112]. Phosphorylation of serine/threonine kinase 1 (Akt), mediates calcium-dependant protein kinase II (CaKMII), and increases phosphorylation of eNOS, thereby increasing NO production and vasodilation [93,113]. Adenosine monophosphate-activated protein kinase (AMPK) increases muscular uptake of glucose, thereby reducing adipose cytokine release, inducing anti-inflammatory effects [83,114]. TRPV1 also regulates transcription of nuclear factor kappa B (NF-ĸB), therefore assisting modulation of cytokine transcription factors and reducing inflammation [94]. Abbreviations: ↑: increased; ↓: decreased.

## 5. The Effects of Capsaicin on Cognition and Cerebrovascular Function

### 5.1. Cognition in Animal Studies

Capsaicin’s action in reducing obesity, oxidative stress, and inflammation and im-proving cardiovascular function in animals has been previously described [83,86,105]. Obesity increases blood triglycerides, free fatty acids and low-density lipoprotein (LDL), contributing to eNOS dysregulation, vascular remodelling, and atherosclerosis, which is the leading cause of cardiovascular and cerebrovascular disease [83,93,106].

Tau proteins are concentrated in the central nervous system and are involved in microtubule assembly [115]. Abnormal tau proteins can increase phosphorylation and decrease microtubule binding, forming amyloid oligomers (such as amyloid-beta, Aβ) or aggregated deposits, which impair brain function by reducing intra- and inter-neuronal signalling, leading to cognitive decline [116]. Intraperitoneal administration of 1 mg/kg of capsaicin for two weeks restored Aβ-induced memory deficits via improved hippocampal synaptic function in C57BL/6 mice. This possibly occurred because it increased the expression of the neuroprotective protein postsynaptic density protein 95 (PSD95), which is often reduced with AD [117]. The increase in PSD95 improved spatial learning in adult C57B1/6 mice, as its primary role is to maintain synaptic plasticity and promote inter-neuronal signalling. Balleza-Tapia et al. [24] also found that hippocampal homogenate with tissue-bath perfusion of capsaicin significantly reduced levels of Aβ and tau protein via activation of TRPV1 in mice. Intraperitoneal administration of 1 mg/kg of capsaicin in an AD mice model upregulated TRPV1, alleviated AD-type pathologies and improved spatial learning and memory [118]. Shiri et al. [23] found that a 10 mg/kg single dose of capsaicin given intraperitoneally improved cognitive performance via TRPV1 in rats, as measured by passive avoidance learning tests. Pegorini et al. [22] found a single dose of capsaicin between 0.2 and 0.6 mg/kg capsaicin, administered via sub-cutaneous injection, was neuroprotective. The authors reported that capsaicin increased the survival rate of CA1 neurons in the hippocampus seven days post-injection in male Mongolian gerbils. However, low-dose capsaicin (0.1 mg/kg) did not affect cognition. Abdel-Salam et al. [25] reported that either 25 mg or 50 mg/kg/day capsicum extract (1.2% capsaicin) given for 30 days improved memory performance and increased central NO concentration, as well as reduced markers of oxidative stress, inflammation and neurodegeneration in a rat model of AD. They also reported that 50 mg/kg/day of capsicum extract reduced oxidative stress and inflammation in non-AD rats compared to the control group.

Cholinesterase enzymatically breaks down acetylcholine, which is a powerful cholinergic vasodilatory neurotransmitter that declines with ageing [119,120]. Cholinesterase is increased in AD, further reducing acetylcholine. Therefore, cholinesterase inhibitors are a current first-line treatment of AD pathologies [121,122]. Rajashri et al. [123] found 13 days of dietary chilli oleoresin containing capsaicin (50 mg/kg; 1.9% *w*/*w*) given with scopolamine (an anticholinergic), reduced acetylcholinesterase (AChE) by 50%. Scopolamine alone decreased AchE, however, in the absence of a capsaicin only arm, it is unclear whether this was a result of capsaicin promoting scopolamine’s actions. Viayna et al. [124] also reported that capsaicin (2 mg/kg intraperitoneally, three times per week for four weeks), scaffolded with the cholinesterase inhibitor huprine Y significantly reduced the Aβ42/Aβ40 ratio in the hippocampus. The reduction in this ratio improved spatial learning and memory and decreased neuroinflammation and hippocampal oxidative stress in APP/PS1 mice. Shalaby et al. [125] reported that 47 days of intragastric infusion of capsaicin at a dose of 10 mg/kg in mice significantly ameliorated Aβ1-42 peptide and tau proteins in the hippocampus, abolishing behavioural impairments. A capsaicin-rich diet (0.01%, approximately 30 mg/kg capsaicin) for six months improved spatial learning and memory consolidation in an AD mice model [27]. This showed that chronic capsaicin intake reduced the total Aβ burden by 32.3% and significantly attenuated tau hyperphosphorylation in both the neocortex and hippocampus. Wang, J. et al. [27] also found that capsaicin significantly reduced proinflammatory cytokines IL-6 and TNF-α, and improved the expression of neuroprotective post-synaptic proteins, such as PSD95, thereby ameliorating neuroinflammation.

cAMP-response-element-binding-protein (CREB), a transcription factor critical in maintaining spatial and long-term memory, is downregulated in AD [126]. Increased phosphorylation of CREB is linked to the binding of calcitonin gene-related peptide (CGRP), a potent vasodilator [127,128]. Intragastric administration of a single dose of capsaicin (10 mg/kg) increased the expression of CREB [129]. Furthermore, 1 m/kg subcutaneous administration of capsaicin for eight days increased CGRP tissue levels in the hippocampus [130]. Therefore, capsaicin increased spatial memory and cognitive performance [129,130]. This action was supported by Bashiri et al. [131], who found that intrahippocampal capsaicin injections (0.05; 0.1; or 0.3 µg/rat) augmented mRNA expression of cyclic adenosine monophosphate (cAMP) and TRPV1 in the CA1 area of the hippocampus, improving memory in rats with biliary cirrhosis.

Avraham et al. [132] found improvements in neurological scores and cognition up to 14 days following a single dose intraperitoneal injection of capsaicin (1.25 µg/kg) in female Sabra mice with hepatic failure. Further, these effects were reversed with the application of a TRPV1 antagonist, confirming that the observed effects of capsaicin were vanilloid mediated [132].

Together these results demonstrate capsaicin’s ability to reduce oxidative stress and inflammation centrally and systemically via TRPV1, thereby alleviating cognitive deficits in animal models of disease. As these findings are all associated with reduced inflammation, it is logical to conclude that the underlying mechanisms of ED could also influence cognitive performance, and these findings could therefore translate to the human cerebrovasculature.

### 5.2. Cognition in Human Studies

Only one study assessed the effects of capsaicin on cognition. Liu et al. [133] assessed the chilli pepper consumption of 338 community-dwelling people (>40 years old) from Chongqing, China, using a self-reported food frequency questionnaire. Cognition was measured using a Chinese version of the Mini-Mental State Examination (MMSE). A capsaicin-rich diet was positively correlated with significantly higher MMSE scores. However, those who consumed chilli daily were younger than those who self-reported weekly chilli consumption. This was attributed to the social phenomenon that older people preferred bland diets. Although limitations of this study also include the inability to quantify self-reported chilli consumption, this demonstrates the possibility that capsaicin could influence human brain health, and that further studies are required to test the effects of chronic chilli consumption on cognition.

### 5.3. Cerebrovascular Function in Animals and Humans

Limited studies have investigated the effect of capsaicin on cerebrovascular function. In vitro studies of feline MCAs found vasodilation of pial arteries occurred with a lower dose of capsaicin (5 × 10^−8^ M) compared to high dose capsaicin (3 × 10^−7^ M) which had vasoconstrictive outcomes [134]. More recently, Marics et al. [135] tested the dural application of capsaicin on meningeal blood flow in vivo, using a laser Doppler flowmeter positioned over a branch of the middle meningeal artery of rats. CGRP release from the meningeal afferents was elevated in response to both low (10 µM) and high (100 µM) topical capsaicin application on the dura mater. Further, CGRP release was seen to be higher in obese rats than in control rats [135]. This may be due to decreased CGRP receptor sensitivity to CGRP (i.e., increased resistance of the CGRP receptor to CGRP) [136]. Xu et al. [137] reported that six months of dietary capsaicin (0.01% for mice and 0.02% for rats) increased the phosphorylation of eNOS in carotid arteries which was associated with the activation of TRPV1. Marquez-Romero et al. [138] pipetted escalating capsaicin dosages (33, 66, 99, 132, 165 µM) onto filtered paper. A single application was applied for 20 min to the hemi-palate of 30 healthy undergraduate students and CBF was measured using TCD. Capsaicin increased CBF, thus demonstrating its potential to elicit vasodilatation. Given the link between CBF and cognition and the limited studies performed using capsaicin, more studies are needed to determine the effects of capsaicin on cerebrovascular function.

Recently, the synthetic capsaicin analogue, nitro-capsaicin, has been studied for its effect on the brain structure and cognition [139]. Nitro-capsaicin substitutes the OCH_3_ group on capsaicin with NO_2_ and produces less gastro-intestinal irritation compared to capsaicin [139,140]. Jamornwan et al. [140] found that nitro-capsaicin, in both vascular damaged and control microglial cell cultures, suppressed microglial activation, decreasing proinflammatory cytokines, such as TNF-α and IL-6, and enhanced anti-inflammatory factors, such as IL-4 and IL-10. Further, when mice with aberrant e4 apolipoprotein E (ApoE4) genes were given 1 mg/kg/day intraperitoneal capsaicin for 1 month, it reversed impaired lipid metabolism, microglial dysfunction, and other neuronal impairments induced by mutant ApoE4 [141]. This demonstrates the potential of capsaicin and capsaicin analogues to reduce chronic microglial activation, a risk factor of neurodegenerative disease and cognitive decline.

## 6. Capsaicin Summary

Although capsaicin was first isolated in 1876 and has been extensively studied, it is only in recent years that its benefits on endothelial function and cognition have become a research focus. Research in animals demonstrates the beneficial effects of capsaicin as a therapeutic agent in improving cognition (Figure 4). These findings suggest that capsaicin’s intracellular binding to TRPV1 activates pathways that modulate inflammation (systemic and neural), oxidative stress, and improve NO bioavailability. This was particularly demonstrated in AD and obesity mice models. All studies with outcomes on Aβ showed significant improvements in the Aβ or Aβ ratio in the hippocampus with various dosages (1–50 mg/kg) and various dietary applications. Further, capsaicin can cross the blood–brain barrier (BBB) and activate TRPV1, as well as increase its expression [24,131,142]. This capsaicin-induced activation decreases BBB permeability, consequently increasing its integrity [143]. This has been directly related to significant memory and spatial learning improvement in AD-type pathologies [27,117,118,124], and traumatic brain injury [143]. Due to this, capsaicin attenuates the progression of neurological disorders, such as dementia in animals.

Despite the evidence that capsaicin can attenuate cognitive decline in animals, limited human studies have been conducted. Further, no clinical studies have been completed that test the effects of capsaicin on cerebrovascular function or cognition. This is summarised in Table 2. This could be due to the technical difficulties associated with testing cerebrovascular function [43]. This may also be due to the pungency of capsaicin which can cause side effects of gastrointestinal irritation and discomfort with ingestion [83,144]. Pungency-related issues with oral administration need to be overcome to test its effects on cerebrovascular function and cognition in humans. Clinical trials assessing the effects of capsaicin on neuroprotection would be beneficial [26]. Therefore, developing novel vehicles for capsaicin delivery could see capsaicin being a suitable treatment option for the management of chronic diseases.

**Table 2 nutrients-15-01537-t002:** Summary of studies that have investigated the effects of capsaicin on cognition and cerebrovascular function.

Reference	Species and Characteristics	Capsaicin Dosage/Application (Duration)	Outcomes of Capsaicin Treatment
Effects of acute dose capsaicin on cognition in animals			
[143]	Adult male Sprague Dawley rats	125 mg/kg s.c. (3 days)	↓BBB permeability ↓vasogenic oedema formation ↓motor and cognitive deficit ↑free magnesium
[22]	Ischaemia model Male Mongolian gerbils	0.025, 0.05, 0.01, 0.2, 0.6 mg/kg s.c.	↑cognition ↑neuronal activity
[145]	Young male Wistar rats	10 mg/kg i.G.	↑spatial memory ↑neuronal long-term potentiation
[132]	Female Sabra mice	1.25 µg/kg i.p. (single dose)	↑cognitive function ↑neurological score
[130]	C57BL/6 WT mice	1 mg/kg i.v. (8 days)	↑spatial learning ↑hippocampal CGRP ↑IGF-1 expression
[129]	Sprague Dawley rats (Grade II, male)	10 mg/kg i.G. (single dose)	↑cognitive performance ↑hippocampal CREB
[23]	Male Wistar mice	10 mg/kg i.p. (single dose)	↑cognitive performance
[24]	P17-30 C57BL/6 (WT) and TRPV1 KO male mice	Hippocampal slices (tissue bath)	↓Aβ neuronal degradation
[131]	Male Wistar rats	0.5, 0.3, 0.1 µg/rat (intrahippocampal injection)	↑memory ↑TRPV1 ↑cAMP mRNA
[118]	APP23/PS45 double transgenic AD mice	1 mg/kg i.p.(single dose)	↑spatial learning ↑memory ↑TRPV1 upregulation ↓hippocampal neurotic plaques
Effects of chronic dose capsaicin in animals			
[117]	Adult C57B1/6 mice	1 mg/kg/day i.p. (2 weeks)	↑spatial learning ↑memory ↑PSD95 expression ↓synapse loss
[25]	Adult male Sprague Dawley rats	AlCl_3_ + 25, 50 mg/kg/day i.p. capsicum extract 50 mg/kg/day i.p. capsicum extract (1.2% capsaicin) (30 days)	↓neuro and systemic inflammation ↓oxidative stress ↓Aβ-peptide accumulation ↓cerebral cortex, substantia nigra and hippocampal neurodegeneration *↑* brain NO concentration and memory
[125]	Adult albino mice	10 mg/kg i.G. (47 days)	↓behavioural impairments ↓Aβ1-42 ↓tau proteins
[123]	Male Wistar rats	Scopolamine + 50 mg/kg oral chilli oleoresin (13 days)	↓Acetylcholinesterase (−50%) ↑locomotion activity ↓escape latency time
[141]	Male, female C57BL/6J littermate ApoE4 mice	1 mg/kg i.p./day (1 month)	↓memory impairment ↓tau pathology ↓neuronal autophagy ↓microglial phagocytosis
[27]	APP/PS1 transgenic mice on C57BL/6 background	0.1% capsaicin-rich chow (approx. 30 mg/kg capsaicin) (6 months)	↑memory ↑spatial learning ↓Aβ plaque density ↓Aβ vessel deposition ↓Aβ42, ↓Aβ40 ↓hyperphosphorylation and tau ↓neuroinflammation ↓neurodegeneration ↓proinflammatory cytokines ↑synapse related proteins
[124]	C47BL6/J mice; APP/PSI mice	2 mg/kg, i.p. (4 weeks)	↑spatial learning ↑memory ↓Aβ42/Aβ40 ratio ↑basal synaptic activity ↓hippocampal oxidative stress ↓hippocampal Neuroinflammation
Effect of capsaicin on animal cerebrovasculature			
[134]	Adult felines	10^−7^–10^−5^ M (tissue bath)	↑vasodilation
[137]	C57BL/6J mice; TRPV1 KO mice; Wistar-Kyoto rats; Stroke-prone, hypertensive male mice	chow + 0.01% dietary capsaicin (0.01% mice; 0.02% rats) (6 months)	↑phosphorylated eNOS ↑eNOS expression Delayed stroke onset TRPV1 function ↑cerebrovascular activity
[135]	Male Sprague Dawley rats	100 nM (low dose), 10 µM (high dose) (in vivo and dural application) (20 weeks)	↑meningeal blood flow ↑CGRP release
Effect of capsaicin on human cerebrovasculature			
[138]	Healthy male adults (n = 30) 21 ± 5 years old	33, 66, 99, 132, 165 µM dose escalation Hemi-palate (20 min)	↑MCA velocity
Effect of capsaicin on cognition in humans			
[133]	Long-term community dwelling adults (n = 338) ≥40 years	Capsaicin dietary intake assessed by food frequency questionnaire	↑cognition Improved Aβ42/Aβ40 ratio

AlCl_3_: Aluminum Chloride; Aβ: beta-amyloid peptide; AD: Alzheimer’s disease; BBB: blood–brain barrier; cAMP: cyclic adenosine monophosphate; conc: concentration; CREB: cAMP-response-element-binding-protein; eNOS: endothelial nitric oxide synthase; CGRP: Calcitonin gene-related peptide; i.G.: intragastric infusion; IGF-I: insulin-growth factor-1; i.p.: intraperitoneally; i.v.: intravenously; KO: knockout; MCA: middle cerebral artery; N: number of participants; NO: nitric oxide; s.c.: subcutaneous injection; PSD95: postsynaptic density protein-95; mRNA: messenger ribonucleic acid; N: number of participants; TRPV1: transient receptor potential vanilloid-1; WT: wild type; ↑: increased; ↓: decreased.

**Figure 4 nutrients-15-01537-f004:**
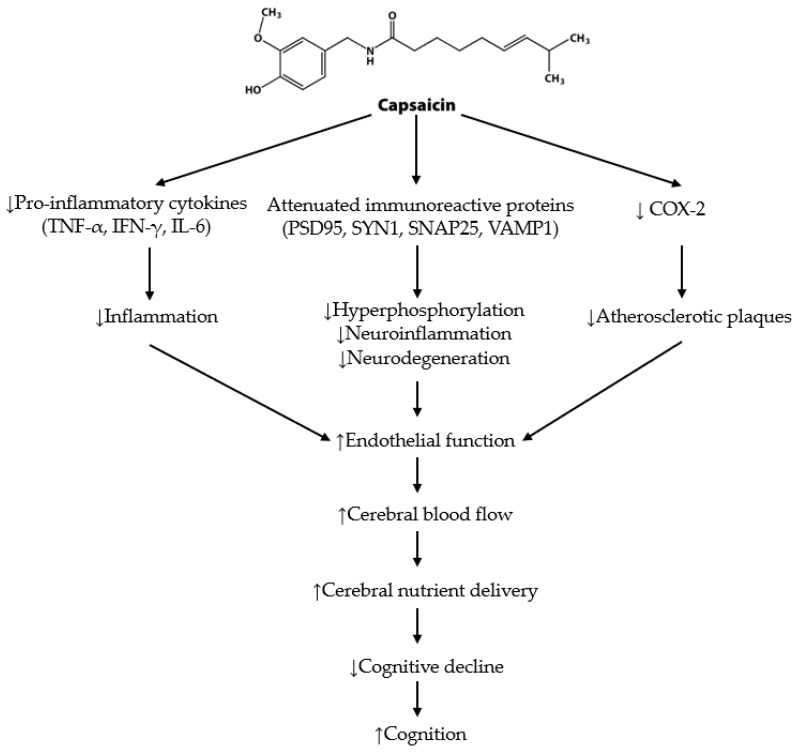
Schematic representation of the neuroprotective effects of capsaicin in animals. This figure demonstrates the capacity of capsaicin to increase CBF, therefore increasing the delivery of nutrients to the brain, a key factor in the prevention of cognitive decline and improved cognition [10,26,43]. Capsaicin decreases pro-inflammatory cytokines, including tumour necrosis factor-alpha (TNF-α), interferon-gamma (IFN-γ) and interleukin-6 (IL-6), reducing inflammation [27,108]. Attenuation of immunoreactive proteins, including postsynaptic density protein 95 (PSD95), synapsin-1 (SYN1), synaptosomal-associated protein 25 (SNAP25) and antivesicle-associated membrane protein 1 (VAMP1), contribute to reducing hyperphosphorylation, neuroinflammation and neurodegeneration [27]. Capsaicin also reduces the action of cyclo-oxygenase (COX-2), which prevents atherosclerotic plaque formation [146,147]. These changes can lead to improved endothelial function, CBF and, in turn, cognition [27,146]. Abbreviations: ↑: increased; ↓: decreased.

## 7. Capsimax

Capsimax is a readily available capsaicin supplement containing 2% capsaicinoids [1.2–1.35% capsaicin, 0.6–0.8% dihydrocapsaicin, and 0.1–0.2% nordihydrocapsaicin]) from 15–25% Capsicum extract, 45–55% sucrose, and 30–35% cellulose gum coatings [148]. Capsimax prolongs release into the small intestine, which therefore overcomes pungency and related issues associated with direct chilli or capsaicin application [148,149,150,151]. OmniActive Health Technologies dissolution studies indicate a range of 60–75% of the total capsaicinoids content of Capsimax are released within four hours and are completely released beyond six hours post-ingestion [149]. A pilot study by Deshpande et al. [148] of incremental dose escalation 2, 4, 6, 8 and 10 mg of capsaicin (i.e., 100, 200, 300, 400 and 500 mg Capsimax) for seven days found all dosages tolerable and safe with no toxic effects reported on vital signs, and metabolic, anthropometric and biochemical markers. As Capsimax is a relatively new product to market, few studies have been performed, particularly in brain health. Studies have focused on obesity, potentially due to this being the reported outcome observed in animal models.

Capsimax has been reported to be a safe and effective weight management strategy, using a dose of 0.84 mg/kg for 52 days in obese C757BL/6J mice [152]. Mariwala et al. [152] found that Capsimax increased thermogenesis and reduced lipogenesis by decreasing biomarkers of adipogenesis, including peroxisome proliferator-activated receptor gamma and fatty-acid binding protein 4. Oxidative stress and inflammation were decreased in male Wistar rats supplemented with 10 mg/kg of Capsimax daily for eight weeks in combination with either exercise [108], a low-fat high-sucrose, or high-fat diet [153]. This was primarily achieved by reduced nuclear factor erythroid-related factor 2, an antioxidant transcription factor, and NF-ĸB. Sahin et al. [153] also reported that Capsimax increased phosphorylation of eNOS in the aorta demonstrating potential for exploration of capsaicin to also improve endothelial function through eNOS phosphorylation.

A single-dose crossover study by Bloomer et al. [149] found 100 mg Capsimax alongside a protein-rich meal increased free fatty acid concentrations at 2 and 2.5 h and glycerol at 4 h post-consumption in exercise-trained, non-smoking adults. Deng et al. [154] and Rigamonti et al. [155] both completed single-dose, single-blinded cross-over studies using 100mg Capsimax. Both studies reported that Capsimax increased resting energy expenditure +7.23% [154] and +16.44% [155] compared to placebo in males and females.

A double-blinded 12-week randomised control trial conducted by Rogers et al. [156] and Urbina et al. [157], each separated groups using either 100 or 200 mg Capsimax or placebo. After 12 weeks, Rogers et al. [156] found that 200 mg of Capsimax reduced fat mass compared to placebo. In contrast, Urbina et al. [157] found no effect on body composition but did report a reduction in waist-to-hip ratio with 100 mg Capsimax at six weeks. A decrease in calorie consumption of 257 kcal per day and increased serum high-density lipoprotein (HDL) was reported in participants given 200 mg Capsimax, compared to 100 mg Capsimax or placebo at 12 weeks. Similarly, a 12-week study comparing 200 mg Capsimax or placebo alongside a calorie-restricted diet reported reductions in body weight, BMI, waist circumference, and fat mass in overweight and obese women of reproductive age [150]. This study reported a high dropout rate of 27%, which was due to the difficulties of compliance with the prescribed restrictive diet rather than side effects of the intervention. As restrictive diets can reduce adiposity [158], the outcomes found by Manca et al. [150] could therefore be a result of the caloric restriction in this intervention, rather than Capsimax supplementation.

These results are summarised in Table 3 and indicate the potential for Capsimax to reduce the oxidative stress and chronic low-grade systemic inflammation that is associated with increased adiposity. As greater inflammation, oxidative stress and adiposity are all factors that influence endothelial function, the reduction of these could lead to improved endothelial function. This may have a flow on effect whereby increased endothelial function improves blood flow and, potentially, CBF and cognition. Although studies have explored the ability of Capsimax to reduce adiposity, a significant risk factor associated with cognitive decline, no clinical trials have been conducted testing the effects of Capsimax on cerebrovascular function or cognition. Given the evidence that capsaicin can reduce cognitive deficits in animal models, dementia research should aim to explore the effects of Capsimax as a potential prevention of cognitive decline.

## 8. Summary

Dementia is a global health care challenge and is characterised by impaired cognition, which leads to a reduced QoL. The brain relies on a constant CBF to preserve cognition. NVC and autoregulation are the complex mechanisms that ensure homeostatic CBF during increased neuronal metabolism and environmental and physical stimuli. If changes in CBF are dysregulated, structural and functional changes occur in the brain, leading to cognitive decline.

Ageing and obesity have synergistic detrimental effects on endothelial function due to hormonal changes, oxidative stress and chronic low-grade systemic inflammation. This causes an uncoupling of eNOS, reducing NO production and increasing oxidative stress, resulting in endothelial dysfunction. Endothelial dysfunction is the main contributing factor of cardiovascular disease, cerebrovascular dysfunction and reduced CBF, thus leading to cognitive decline.

Capsaicin is a pungent molecule found abundantly in chillies and has demonstrated beneficial effects on weight management, oxidative stress and chronic low-grade systemic inflammation. Capsaicin-induced activation of TRPV1 can regulate inflammation and oxidative stress by increasing NO production, thus improving endothelial function. Further, this review indicated that acute and chronic capsaicin treatment attenuated AD-type pathologies and improved cognition in animals. However, studies adequately assessing the effects of capsaicin on cerebrovascular function, and cognition in humans do not exist. Pungency-related side effects, such as gastrointestinal irritation, make determining the effects of capsaicin in humans difficult to assess.

## 9. Conclusions

This review outlines how Capsimax, a capsaicin supplement, has been demonstrated as a novel, safe and tolerable capsaicin supplement that reduces side effects such as gastrointestinal distress associated with capsaicin intake. Capsimax has been tested in both animals and human clinical trials and reduced adiposity via increased thermogenesis and reduced caloric consumption. Capsimax also reduced chronic low-grade systemic inflammation and oxidative stress in animals, as well as improved endothelial function in rats, all of which are hallmarks associated with cognitive decline beyond that of normal ageing.

This review provides justification for adopting Capsimax in clinical trials as an effective, non-pharmaceutical intervention to counteract the decline of cerebrovascular function and cognition, particularly in obesity and ageing. Considering the lack of current preventive strategies to counteract cognitive ageing, capsaicin may present a natural treatment to counteract cognitive decline. Future prospective studies are required to see whether the attenuation of cognitive decline by capsaicin can translate to reduced risk of disease such as dementia and Alzheimer’s in humans.

## Figures and Tables

**Figure 1 nutrients-15-01537-f001:**
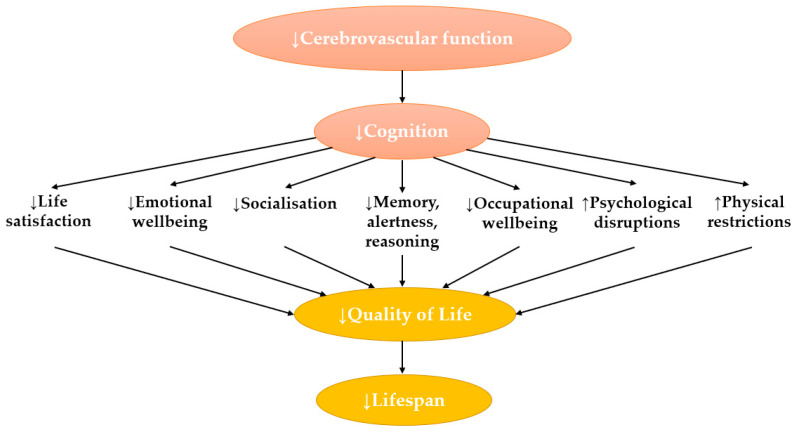
The outcomes of reduced cerebrovascular function leading to reduced quality of life. Lowered cerebrovascular function influences cerebral hypoperfusion, which causes structural and functional changes leading to reduced cognition. Reduction of cognition affects multiple regions of the brain, which influence aspects that contribute to quality of life. This includes psychological, physical, social, and occupational wellbeing. Lowered quality of life can reduce life expectancy [5]. Abbreviations: ↑: increased; ↓: decreased.

**Table 3 nutrients-15-01537-t003:** Review of Capsimax intervention studies.

Reference	Study Design	Species/Participants/Sex	Study Duration	Capsaicin Dosage	Primary Outcome/s
Effects of Capsimax in animal studies					
[152]	Random allocation	C57BL/6J male mice (n = 24)	52 days	0.84 mg/kg/day + high fat diet	↑thermogenesis ↓body mass ↓lipogenesis
[153]	Random allocation	Male Wistar rats (n = 42)	8 weeks	10 mg/kg/day	↓Inflammatory markers ↓body mass
[108]	Random allocation	Male albino Wistar rats (n = 28)	8 weeks	10 mg/kg/day	↑antioxidants ↓inflammation ↑time to exhaustion
Effects of Capsimax in human studies					
[154]	Single-blind, crossover	Healthy adults (n = 40) 17F/23M	3 h	100 mg single dose	↑resting energy expenditure
[149]	Double-blind, crossover, randomised control trial	Healthy adults (n = 20) 10F/10M	3 h	100 mg single dose	↑free fatty acids ↑glycerol
[155]	Single blind, crossover, randomised control trial	Obese, hospitalised 15–34 years in weight reduction program (n = 10) 4F/6M	6 h	100 mg single dose	↑resting energy expenditure ↑satiety ↓hunger
[148]	Open-label, dose-finding, adaptive study	Healthy overweight 25–55 years women (n = 12)	6 weeks	100 mg/day +100 mg/day weekly escalating until 500 mg/day/week	Tolerable up to 500 mg/day No adverse events No dropouts
[150]	Randomised control trial	Overweight or obese 18–51 years women (n = 61)	12 weeks	200 mg/day + diet restriction	↓fat mass ↓body mass ↓waist circumference ↑free fat mass
[156]	Parallel double-blind, randomised control trial	Heathy 18–56 years (n = 77) 47F; 30M	12 weeks	100 mg/day 200 mg/day	↓fat mass (−6.68%) ↓body mass (−5.91%)
[157]	Double-blind, randomised control trial	Healthy adults (n = 77)	12 weeks	100 mg/day 200 mg/day	↓kJ intake ↓waist-to-hip ratio ↓appetite ↓HDL cholesterol

BMI: body mass index; eNOS: endothelial nitric oxide synthase; F: female; HDL: high-density lipoprotein; M: male; N: number of participants; ND: normal diet; NF-ĸB: nuclear factor kappa-light-chain enhancer of activated B cells; ↑: increased; ↓: decreased.

## Data Availability

Not applicable.

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
