# Peer review of "Capsaicin: A Potential Treatment to Improve Cerebrovascular Function and Cognition in Obesity and Ageing"

_nutrients, 2023, doi:10.3390/nu15061537_

Round 1

Reviewer 1 Report

The present paper enlightens the main biological properties of capsaicin and analogues to improve cerebrovascular function, cognition and related cognitive behavior.

 The paper is well organised and written. Each section is properly described. Figures and Tables are informative and clear.

 My only inputs are as follows:

 - Add to ref 18 the citation, as it offers a wider view of AD multifactorial nature

 L. Blaikie, G. Kay, P. Kong Thoo Lin, Current and emerging therapeutic targets of alzheimer’s disease for the design of multi-target directed ligands, Med. chem. commun 10 (2019) 2052–2072, doi: 10.1039/c9md00337a.

 - Add to ref 21-23 and describe this very recent paper.

 Abdel-Salam OME, El-Sayed El-Shamarka M, Youness ER, Shaffie N. Protective effect of hot peppers against amyloid β peptide and brain injury in AlCl3-induced Alzheimer's disease in rats. Iran J Basic Med Sci. 2023 Mar;26(3):335-342. doi: 10.22038/IJBMS.2022.67871.14845.

 - Add to ref 97 the more recent paper where the ability of capsaicin to rescue lipid metabolic impairments and reverse ApoE4-induced microglial immune dysfunction and neuronal autophagy impairment is reported.

  Wang C, Lu J, Sha X, Qiu Y, Chen H, Yu Z. TRPV1 regulates ApoE4-disrupted intracellular lipid homeostasis and decreases synaptic phagocytosis by microglia. Exp Mol Med. 2023 Feb;55(2):347-363. doi: 10.1038/s12276-023-00935-z.

 - The role of microglia in aging and neurodegeneration and the role of capsaicin derivatives have to be discussed. Take into consideration the following papers.

 Pinosanu LR, Capitanescu B, Glavan D, Godeanu S, Cadenas IFN, Doeppner TR, Hermann DM, Balseanu AT, Bogdan C, Popa-Wagner A. Neuroglia Cells Transcriptomic in Brain Development, Aging and Neurodegenerative Diseases. Aging Dis. 2023 Feb 1;14(1):63-83. doi: 10.14336/AD.2022.0621.

 Villa V, Thellung S, Bajetto A, Gatta E, Robello M, Novelli F, Tasso B, Tonelli M, Florio T. Novel celecoxib analogues inhibit glial production of prostaglandin E2, nitric oxide, and oxygen radicals reverting the neuroinflammatory responses induced by misfolded prion protein fragment 90-231 or lipopolysaccharide. Pharmacol Res. 2016 Nov;113(Pt A):500-514. doi: 10.1016/j.phrs.2016.09.010.

 Jamornwan S, Chokpanuwat T, Uppakara K, Laorob T, Wichai U, Ketsawatsomkron P, Saengsawang W. Nitro Capsaicin Suppressed Microglial Activation and TNF-α-Induced Brain Microvascular Endothelial Cell Damage. Biomedicines. 2022 Oct 23;10(11):2680. doi: 10.3390/biomedicines10112680.

Author Response

Reviewer #1: Review

The present paper enlightens the main biological properties of capsaicin and analogues to improve cerebrovascular function, cognition and related cognitive behavior.

 The paper is well organised and written. Each section is properly described. Figures and Tables are informative and clear.

We sincerely thank the reviewer for their comments and suggestions. Please see our point-by-point responses and corrections below (or in the attached Word Document). The line numbers refer to the Redlined Article File.

 My only inputs are as follows:

 - Add to ref 18 the citation, as it offers a wider view of AD multifactorial nature

  1. Blaikie, G. Kay, P. Kong Thoo Lin, Current and emerging therapeutic targets of alzheimer’s disease for the design of multi-target directed ligands, Med. chem. commun 10 (2019) 2052–2072, doi: 10.1039/c9md00337a.

This article has now been included (Ref #21) as it supports the findings and need for multi-functional treatments to address cognitive decline (Lines 56-58; 61).

 - Add to ref 21-23 and describe this very recent paper.

 Abdel-Salam OME, El-Sayed El-Shamarka M, Youness ER, Shaffie N. Protective effect of hot peppers against amyloid β peptide and brain injury in AlCl3-induced Alzheimer's disease in rats. Iran J Basic Med Sci. 2023 Mar;26(3):335-342. doi: 10.22038/IJBMS.2022.67871.14845.

This article is very recent and published after submission of this paper. We have now included this in Table 2 (Ref #25) and discussed the findings within the paper (Lines 298-303).

 - Add to ref 97 the more recent paper where the ability of capsaicin to rescue lipid metabolic impairments and reverse ApoE4-induced microglial immune dysfunction and neuronal autophagy impairment is reported.

  Wang C, Lu J, Sha X, Qiu Y, Chen H, Yu Z. TRPV1 regulates ApoE4-disrupted intracellular lipid homeostasis and decreases synaptic phagocytosis by microglia. Exp Mol Med. 2023 Feb;55(2):347-363. doi: 10.1038/s12276-023-00935-z.

This reference has now been incorporated into Table 2 (Ref #141). This paper has also been summarised in the body of the article (Lines 386-389).

 - The role of microglia in aging and neurodegeneration and the role of capsaicin derivatives have to be discussed. Take into consideration the following papers.

The role of microglia in aging and neurodegeneration and the role of capsaicin derivatives has now been included (Refs #62-66) (Lines 150-154).

Pinosanu LR, Capitanescu B, Glavan D, Godeanu S, Cadenas IFN, Doeppner TR, Hermann DM, Balseanu AT, Bogdan C, Popa-Wagner A. Neuroglia Cells Transcriptomic in Brain Development, Aging and Neurodegenerative Diseases. Aging Dis. 2023 Feb 1;14(1):63-83. doi: 10.14336/AD.2022.0621.

This article has been reviewed and included (Ref #62) and used to support the introduction to microglia (Lines 150-151).

Villa V, Thellung S, Bajetto A, Gatta E, Robello M, Novelli F, Tasso B, Tonelli M, Florio T. Novel celecoxib analogues inhibit glial production of prostaglandin E2, nitric oxide, and oxygen radicals reverting the neuroinflammatory responses induced by misfolded prion protein fragment 90-231 or lipopolysaccharide. Pharmacol Res. 2016 Nov;113(Pt A):500-514. doi: 10.1016/j.phrs.2016.09.010.

This article has been reviewed and included (Ref #65) and used to support the introduction to microglia (Lines 152-154).

Jamornwan S, Chokpanuwat T, Uppakara K, Laorob T, Wichai U, Ketsawatsomkron P, Saengsawang W. Nitro Capsaicin Suppressed Microglial Activation and TNF-α-Induced Brain Microvascular Endothelial Cell Damage. Biomedicines. 2022 Oct 23;10(11):2680. doi: 10.3390/biomedicines10112680.

This article has been reviewed and included (Ref #140; Lines 380-389).

Reviewer 2 Report

See: attached file

Author Response

Reviewer #2: Review

I have read the above review with an interest. The authors aimed to clearly present the role and potential use of capsaicin in the improvement of cognition and cerebrovascular function.

However, there are a few of weaknesses, which in my opinion should be corrected.

We sincerely thank the reviewer for their comments and suggestions. Please see our point-by-point responses and corrections below (or in the attached Word Document). The line numbers refer to the Redlined Article File.

General concept comments:

Line 62-67

„Therefore, the aims of this review are to:

1) provide a brief overview of cognition and cerebrovascular function;

2) examine the mechanisms underlying how increased adiposity and ageing lead to reduced cerebrovascular function and cognition;

3) examine how capsaicin attenuates the effects of ageing and obesity on decreased cerebrovascular function and cognition; and

4) provide a summary of potential health benefits that are associated with Capsimax, a capsaicin supplement associated with reduced gastrointestinal irritation.”

Excellent construction of this review! However, each part is definitely too long. Please concentrate on the key points you want to share with the public. Try to indicate more areas of interest where capsaicin can be possibly used.

This feedback has been taken as the assumption that it refers to reducing the detail within the review, rather than reducing the wording of the review aims. Sections of this review have been edited to address this comment. This includes information in both figure explanations and corresponding sections to not include repetitive information. We should also note that the other reviewer of this article has requested additional content be added in specific sections and provided the following comment:

“The paper is well organised and written. Each section is properly described. Figures and Tables are informative and clear.”

Hence, we have ensured that we balance the content based on the comments provided by both reviewers, being respectful of the suggestions listed by the reviewer here on what could and should be removed. Changes made are listed below and as part of the response to corresponding comments further below.

Line 68: The words ‘that are’ removed.

Line 111: The words ‘such as the tumour necrosis factor-α (TNF-α) production pathway’ removed.

Lines 177-180: We have reduced this section by three lines from the original manuscript. This now reads more concise and ensure repetition is avoided.  “A meta-analysis of seven observational studies demonstrated that obesity increased the risk of dementia between 32 – 45% due to increased concentrations of inflammatory markers.”

Lines 249-251: Information has been added to address additional areas where capsaicin can be possibly used. “This may also indicate capsaicin’s application as a resolution to other inflammatory diseases such as autoimmune diseases, gastrointestinal disorders, and other haemodynamic conditions, such as atherosclerosis.” We should also note that it is out of the scope of this review to assess the use of capsaicin in other chronic diseases in detail, as this has been reviewed elsewhere or requires further research.

Specific comments:

Line 34-36

„Dementia is a major cause of functional disability, with a substantial social and economic burden estimated to cost AUS$2.7 trillion globally by 2030 [1,2].”

Please provide current data concerning economic costs of dementia and express it also in US$/EUR.

This information has been updated to reflect US$2.8 trillion by 2030 (Lines 35-36). Additional reference included for updated information. World Health Organization. Dementia. 2022. Available online: https://www.who.int/news-room/fact-sheets/detail/dementia.  

Line 156-182

In my opinion, some information is missing in this part: a few sentences on adipokines. How many molecules have been described so far? Etc. Please do not repeat information from the description of Figure 2.

Additional information has now been included regarding adipokines (Lines 167-173).

Information within the figure descriptions and within the text has also been revised to remove repetitive information.

Lines 191-192: This content has been reduced by nearly 2 lines, with all specific markers being removed to ensure repetition is avoided with respect to the content in Figure 2. This also applies for lines 197-199, which has also been further reduced by 1 line, thus ensuring repetitive descriptors are removed.  

Line 227-237

Please do not repeat information from the description of Figure 3.

Information within the figure descriptions and within the text has been revised to remove repetitive information.

Lines 216-218: This content has been reduced by nearly 4 lines, with all specific markers and descriptors being removed to ensure repetition is avoided with respect to the content in Figure 4.

Line 236: We have removed content to ensure that this is more succinctly summarised to avoid repetition.

Ref 8 & 44 - inappropriate self-citations by author.

We have removed all self-citations, other than Ref #85 (Line 202) as this is the only work thus far that describes and summates the evidence from the current literature as evidenced by the amount of citations this paper has received thus far.

Line 125: Replaced self-citation with Ref #32 Toth, P., et al., Functional vascular contributions to cognitive impairment and dementia: mechanisms and consequences of cerebral autoregulatory dysfunction, endothelial impairment, and neurovascular uncoupling in aging. American Journal of Physiology: Heart and Circulatory Physiology, 2017. 312(1): p. H1-H20.

Round 2

Reviewer 2 Report

Congratulations! I accept the revised manuscript. I do appreciate the effort to satisfy both reviewers.